# Off-Policy Evaluation with Out-of-Sample Guarantees

**Sofia Ek**                                                                                      *sofia.ek@it.uu.se*
*Department of Information Technology*
*Uppsala University*

**Dave Zachariah**                                                                        *dave.zachariah@it.uu.se*
*Department of Information Technology*
*Uppsala University*

**Fredrik D. Johansson**                                                          *fredrik.johansson@chalmers.se*
*Department of Computer Science & Engineering*
*Chalmers University of Technology*

**Petre Stoica**                                                                                          *ps@it.uu.se*
*Department of Information Technology*
*Uppsala University*

**Reviewed on OpenReview:** *https://openreview.net/forum?id=XnYtGPgG9p*

## Abstract

We consider the problem of evaluating the performance of a decision policy using past observational data. The outcome of a policy is measured in terms of a loss (aka. disutility or negative reward) and the main problem is making valid inferences about its out-of-sample loss when the past data was observed under a different and possibly unknown policy. Using a sample-splitting method, we show that it is possible to draw such inferences with finite-sample coverage guarantees about the entire loss distribution, rather than just its mean. Importantly, the method takes into account model misspecifications of the past policy – including unmeasured confounding. The evaluation method can be used to certify the performance of a policy using observational data under a specified range of credible model assumptions.

## 1 Introduction

In this work, we are interested in evaluating the performance of a decision policy, denoted $\pi$, which chooses an action from a discrete action set. Each action $A$ is taken in a context with observable covariates $X$ and incurs a real-valued loss $L$ (aka. disutility or negative reward). Such policies are considered, for example, in contextual bandit problems and precision medicine (Langford & Zhang, 2007; Qian & Murphy, 2011; Lattimore & Szepesvári, 2020; Tsiatis et al., 2019). For instance, $A$ may be one of several treatment options for a patient with observable characteristics $X$, and $L$ measures the severity of the outcome.

A target policy $\pi$ can be evaluated using experimental data obtained from trials. Such experiments are, however, often costly to perform and may lead to rather small sample sizes in, e.g., clinical settings. Moreover, in safety-critical applications, it is often unethical to test new policies without severe restrictions on the trial population as well as the target policy. A more fundamental inferential problem, however, is the lack of 'external' validity, i.e., the limited ability to extrapolate from the trial population to the intended target population may lead to invalid inferences (Westreich, 2019; Manski, 2019). The main alternative is *off-policy* evaluation, i.e., using observational data from a past decision process to infer the performance of the target policy. For this to be valid one needs to *assume* that the past process is modeled without systematic errors, i.e, no unmeasured confounding and using well-specified models. The credibility of these assumptions, therefore, determines the 'internal' validity of inferences about $\pi$ from observational data (Manski, 2003).

Inferences that lack validity are particularly serious when evaluating $\pi$ in decision processes that are costly or safety-critical. In such cases, even inferences that are asymptotically valid with increasing sample size may not be adequate. Moreover, when the resulting distribution of losses is skewed or is widely dispersed, its tails are important to evaluate. Then inferring the expected loss $\mathbb{E}_\pi[L]$, as is commonly done, provides a very limited evaluation of $\pi$ (Wang et al., 2018). For instance, the average loss in a population may be small but the tail losses can be unacceptably large. In such applications, we are more concerned with providing valid certifications of the overall performance (see Figure 1a), rather than inferences of a single distributional parameter.

In this paper, we propose a method for evaluating a specified target policy using observational data that

- provides finite-sample coverage guarantees for the out-of-sample loss,

- evaluates the entire loss distribution instead of, e.g., its expected value,

- and takes model misspecification, including unmeasured confounding, into account.

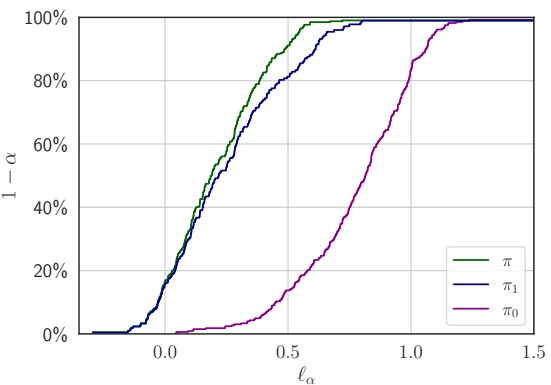
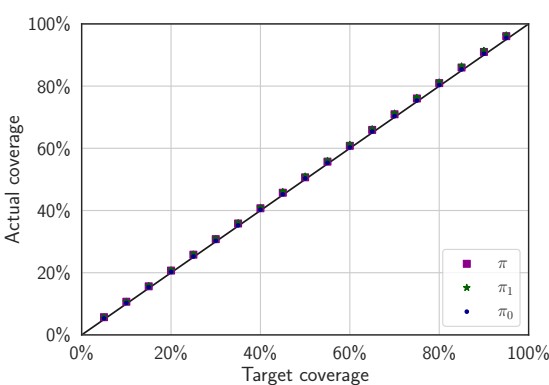

(a) Evaluation of out-of-sample loss $L$.

(b) Evaluation of coverage of limit curves.

Figure 1: Evaluating out-of-sample losses under target policy with binary decisions $A \in \{0, 1\}$. Policies $\pi_0$ and $\pi_1$ correspond to 'treat none' ($A \equiv 0$) and 'treat all' ($A \equiv 1$), respectively, while $\pi$ denotes a policy that adapts to context $X$. (For more details, see the example in Section 5.1 with $p_1$.) (a) Each curve certifies that a new loss $L$ falls below the limit $\ell_\alpha$ with a probability of at least $1 - \alpha$ using $n = 1000$ data points. The certified performance of policy $\pi$ dominates those of the alternative policies. (b) Evaluation of actual coverage, that is, the probability of $L \leq \ell_\alpha$, versus target coverage $1 - \alpha$.

## 2 Problem formulation

We consider a *target policy* $\pi$ for deciding an action $A$ in different contexts, which are described by observed and unobserved covariates $X$ and $U$, respectively. The policy corresponds to a distribution $p_\pi(A|X)$, and can be either deterministic or random. Our aim is to evaluate the losses $L$ that result from applying any given $\pi$. Each instance of contextual covariates, action, and loss, i.e., $(X, U, A, L)$, is drawn independently from a target distribution $p_\pi(X, U, A, L)$. At our disposal is an *observational* data set

$$\mathcal{D} = \big((X_i, A_i, L_i)\big)_{i=1}^n, \tag{1}$$

and our goal is to use it to characterize the *out-of-sample loss* $L_{n+1}$. Note that it was collected under a possibly *different* policy than $\pi$. If we continue with the example of patients, then $L_{n+1}$ would quantify the severity of the outcome for a new future patient that is unobserved at the time of our evaluation. Specifically, for any miscoverage level $\alpha \in (0, 1)$, we seek an informative limit $\ell_\alpha(\mathcal{D})$ on the loss such that

$$\mathbb{P}_\pi\Big\{L_{n+1} \leq \ell_\alpha(\mathcal{D})\Big\} \geq 1 - \alpha. \tag{2}$$

In other words, $\ell_\alpha(\mathcal{D})$ as a function of $\alpha$ yields a finite-sample performance certification of $\pi$ as is illustrated in Figure 1a. Unlike a single point estimate, the limit curve characterizes the distribution of losses under $\pi$. For instance, consider a patient population where $(X, U)$ denote its covariates, and $A$ is its received treatment with an outcome loss $L$. Then across all coverage levels, (2) ensures that the treatment of a future patient will incur a loss no greater than $\ell_\alpha(\mathcal{D})$ under policy $\pi$ with confidence $1 - \alpha$. Figure 1b shows the probability that a limit $\ell_\alpha(\mathcal{D})$, proposed below, bounds the future loss $L_{n+1}$ across all values of $1 - \alpha$. Note that limit curves can be constructed for several alternative target policies, which enables us to compare and focus on the parts of their loss distributions that are most relevant to the specific decision problem.

The causal structure of this decision process is illustrated in Figure 2a. The target distribution admits a causal factorization

$$p_\pi(X, U, A, L) = p(X, U)\, p_\pi(A|X)\, p(L|A, X, U), \tag{3}$$

where $p(X, U)$ and $p(L|A, X, U)$ are *unknown*. The central challenge in *off-policy* evaluation of $\pi$ is that (1) is not sampled from (3) but from a shifted *training* distribution which admits a causal factorization

$$p(X, U, A, L) = p(X, U)\, p(A|X, U)\, p(L|A, X, U), \tag{4}$$

where $p(A|X, U)$ characterizes a *past policy* (aka. behavioral policy) that may differ from $\pi$. The causal structure corresponding to (4) is illustrated in Figure 2b. If the past policy were known, it would be possible to adjust for the distribution shift from training to target distribution using the ratio

$$\frac{p_\pi(X, U, A, L)}{p(X, U, A, L)} \equiv \frac{p_\pi(A|X)}{p(A|X, U)}. \tag{5}$$

This is feasible in certain problems with fully automated decision-making, such as online recommendation systems, where the past policy is designed using observable covariates only, i.e., $p(A|X, U) \equiv p(A|X)$. In more general problems, however, we have only a *nominal* model of the past policy $\widehat{p}(A|X)$ (aka. propensity model), typically estimated from prior observable data. The nominal model may therefore diverge from $p(A|X, U)$ due to various modelling errors that persist even in the large-sample scenario: model misspecification and unmeasured confounding via $U$ (Peters et al., 2017; Westreich, 2019). Here we follow the marginal sensitivity methodology of Tan (2006) and characterize the model divergence with respect to the odds of taking action $A$. That is, the nominal odds diverge from the unknown odds by some bounded factor $\Gamma \geq 1$ as follows:

$$\frac{1}{\Gamma} \leq \underbrace{\frac{p(A|X, U)}{1 - p(A|X, U)}}_{\text{unknown odds}} \Big/ \underbrace{\frac{\widehat{p}(A|X)}{1 - \widehat{p}(A|X)}}_{\text{nominal odds}} \leq \Gamma, \tag{6}$$

for all discrete actions $A$ and almost surely all $X, U$. When the bound equals $\Gamma = 1$, the nominal model is perfectly specified and there is no unmeasured confounding. In general, we consider all cases where the nominal odds diverge by *at most* a factor $\Gamma$. Note that (6) can accommodate all possible sources of errors in the nominal model $\widehat{p}(A|X)$, including model misspecification as well as finite-sample errors.

In summary, the problem we consider is to construct a limit $\ell_\alpha(\mathcal{D})$ for target policy $\pi$ using a nominal model $\widehat{p}(A|X)$ and $\Gamma$. The resulting limit should satisfy the finite-sample guarantee (2) for all miscoverage levels $\alpha$, and thereby certify the target policy performance for any specified bound $\Gamma$ in (6). This enables a *robust* evaluation of target policies using observational data since it can be performed across a range of credible odds divergence bounds $\Gamma$ as we will illustrate in the numerical experiments in Section 5.

By increasing the divergence bound $\Gamma$, the credibility of our assumptions on $\widehat{p}(A|X)$ increases, but the informativeness of inferences about $L_{n+1}$ decrease, cf. Manski (2003). Therefore we advocate to evaluate a policy $\pi$ using several $\Gamma$ in the range $[1, \Gamma_{\max}]$, where $\Gamma_{\max}$ is the maximum value for which the limit curve remains informative in any given problem. The *informativeness* of a valid limit curve can be defined as follows:

$$\text{Informativeness} = 1 - \alpha^*, \text{ where } \alpha^* = \inf\{\alpha : \ell_\alpha(\mathcal{D}) < L_{\max}\}, \tag{7}$$

where $L_{\max}$ is the maximum value of the support of $L$. That is, the highest coverage probability at which we can informatively certify the performance of $\pi$, i.e. the right limit of a curve. Note that the informativeness is a problem-dependent quantity.

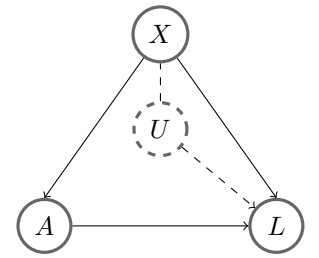

(a) Decision process under target policy

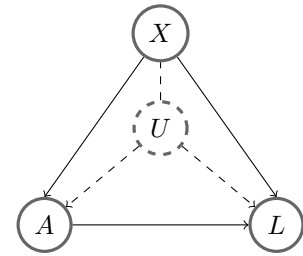

(b) Decision process under past policy

Figure 2: Directed acyclic diagrams representing the causal structure of the decision process under (a) target policy, which yields (3), and (b) past policy, which yields (4). In (b), both contextual covariates $(X, U)$ may joint affect actions $A$ and outcome loss $L$. Thus $U$ gives rise to unmeasured confounding.

## 3 Background

We place the problem considered in this paper and our proposed method within the framework of off-policy evaluation.

**Expected loss:** In the off-policy evaluation literature, the target quantity is often the unknown expected loss $\mathbb{E}_\pi[L]$ of policy $\pi$. A standard estimator of the mean, that dates back to Horvitz & Thompson (1952), is based on inverse propensity weighting:

$$V_{\text{IPW}}(\mathcal{D}) = \frac{1}{n} \sum_{i=1}^n \widehat{w}(X_i, A_i)\, L_i, \tag{8}$$

where $\widehat{w}(X, A) = \frac{p_\pi(A|X)}{\widehat{p}(A|X)}$ is a model of (5), see for instance Rosenbaum & Rubin (1983); Beygelzimer et al. (2009); Qian & Murphy (2011); Zhang et al. (2012); Zhao et al. (2012); Kallus (2018). We note that the estimator in (8) is unbiased when $\Gamma = 1$. An alternative standard estimator is based on regression modeling:

$$V_{\text{RM}}(\mathcal{D}) = \frac{1}{n} \sum_{i=1}^n \sum_{a \in \mathcal{A}} p_\pi(a|X_i)\, \widehat{\ell}(a, X_i), \tag{9}$$

where $\widehat{\ell}(A, X)$ is a model of $\mathbb{E}[L|A, X]$.

The approaches in (8) and (9) have complementary strengths and weaknesses resulting from the challenges of modelling the past policy and the conditional mean of losses, respectively. Even when the models are well-specified, the accuracy of the estimators depends highly on the overlap of covariates $X$ across decisions $A$ in the training data (Oberst et al., 2020; D'Amour et al., 2021). When the overlap is weak, the variance of $V_{\text{IPW}}(\mathcal{D})$ can become excessively large, even when it is unbiased. This can be mitigated by clipping the weights, which in turn causes bias (Rubin, 2001; Kang & Schafer, 2007; Schafer & Kang, 2008; Strehl et al., 2010).

When the models $\widehat{w}(X, A)$ or $\widehat{\ell}(A, X)$ exhibit systematic errors, however, the corresponding estimators in (8) and (9) are biased and may invalidate the evaluation of $\pi$. The 'doubly robust' estimator

$$V_{\text{DR}}(\mathcal{D}) = \frac{1}{n} \sum_{i=1}^n \left( \widehat{w}(X_i, A_i) \left[ L_i - \widehat{\ell}(A_i, X_i) \right] + \sum_{a \in \mathcal{A}} p_\pi(a|X_i)\, \widehat{\ell}(a, X_i) \right),$$

is one way of robustification in the case of *either* model $\widehat{w}(X, A)$ or $\widehat{\ell}(A, X)$ is misspecified. Moreover, it reduces the estimator variance provided $\widehat{\ell}(A, X)$ is sufficiently accurate (Bang & Robins, 2005; Dudík et al., 2011; Rotnitzky et al., 2012).

**Distribution of losses:** When the loss distribution is highly skewed and the tails are long, the use of expected loss to evaluate policies can be inadequate, especially in high-stakes problems. There are alternative

parameters of the loss distribution, described by the cumulative distribution function $F(\ell) = \mathbb{P}_\pi\{L_{n+1} \leq \ell\}$ (cdf), that one can consider in such problems, e.g., a quantile or the conditional value at risk (Wang et al., 2018; Chandak et al., 2021; Huang et al., 2021).

Off-policy evaluation of $\pi$ with respect to some alternative parameter can be achieved using cdf-estimators that are analogous to the mean estimators above, see Huang et al. (2021). In analogy to (8), the inverse propensity weighted cdf-estimator

$$\widehat{F}_{\mathrm{IPW}}(\ell; \mathcal{D}) = \frac{1}{n} \sum_{i=1}^{n} \widehat{w}(X_i, A_i)\, \mathbb{1}(L_i \leq \ell), \tag{10}$$

is point-wise unbiased when $\Gamma = 1$. Similar to (9), the estimator

$$\widehat{F}_{\mathrm{RM}}(\ell; \mathcal{D}) = \frac{1}{n} \sum_{i=1}^{n} \sum_{a \in \mathcal{A}} p_\pi(a|X_i)\, \widehat{c}(\ell; a, X_i),$$

requires a model $\widehat{c}(\ell; a, x)$ of the conditional distribution $\mathbb{P}\{L \leq \ell | A, X\}$. To mitigate against model misspecification that threatens the validity of the evaluation of $\pi$, one can use the 'doubly robust' estimator

$$\widehat{F}_{\mathrm{DR}}(\ell; \mathcal{D}) = \frac{1}{n} \sum_{i=1}^{n} \left( \widehat{w}(X_i, A_i)\Big[\mathbb{1}(L_i \leq \ell) - \widehat{c}(\ell; A_i, X_i)\Big] + \sum_{a \in \mathcal{A}} p_\pi(a|X_i)\, \widehat{c}(\ell; a, X_i) \right),$$

which protects in the case that either model $\widehat{w}(X, A)$ or $\widehat{\ell}(A, X)$ is misspecified. While this estimator is consistent, it is not guaranteed to yield a proper cdf.

In this paper, we are interested in limiting the out-of-sample loss $L_{n+1}$. The $\alpha$-quantile is the smallest $\ell_\alpha$ such that $\mathbb{P}_\pi\{L_{n+1} \leq \ell_\alpha\} \geq 1 - \alpha$. It can be estimated as

$$\ell_\alpha(\mathcal{D}) = \inf\left\{ \ell : \widehat{F}(\ell; \mathcal{D}) \geq 1 - \alpha \right\},$$

using one of the cdf-estimators above. We will use $\widehat{F}_{\mathrm{IPW}}$ as a benchmark below.

**Distribution-free inference:** Derivations of finite-sample valid, nonparametric limits on random variables date back to the works of Wilks (1941); Wald (1943); Scheffe & Tukey (1945). More recently, the related methodology of conformal prediction has focused on developing covariate-specific prediction regions (Vovk et al., 2005; Shafer & Vovk, 2008; Vovk, 2012). See Lei & Wasserman (2014); Lei et al. (2018); Romano et al. (2019) for further developments. Tibshirani et al. (2019) extended the methodology to make it valid also under known covariate shifts. This extended methodology was used to provide context-specific prediction intervals for any given policy $\pi$, which are statistically valid under the assumption that the past policy $p(A|X, U)$ is known (Osama et al., 2020; Taufiq et al., 2022).

The marginal sensitivity methodology developed in Tan (2006) enables us to specify a more credible range of assumptions using (6). This type of methodology was used for robust policy learning in Kallus & Zhou (2021), for robust policy learning in sequential decisions in Namkoong et al. (2020) and for sensitivity analysis of treatment effects in Jin et al. (2023) in the case of unobserved confounding. The present paper considers the overall performance of $\pi$, similar to Huang et al. (2021). However, different from that paper, we focus on ensuring inferences on the out-of-sample losses that are valid even for finite training data and systematic modelling errors – including unobserved confounding – using a sample-splitting technique that leverages results derived in Jin et al. (2023).

## 4 Method

We show that one can limit the out-of-sample losses for $\pi$ under any specified odds divergence bound $\Gamma \geq 1$ for the nominal model in (6), which we assume holds. It follows that (5) is bounded as:

$$\underline{W} \leq \frac{p_\pi(X, U, A, L)}{p(X, U, A, L)} \leq \overline{W}, \tag{11}$$

where the bounds equal

$$\underline{W} = p_\pi(A|X) \cdot \left[1 + \Gamma^{-1}\big(\widehat{p}(A|X)^{-1} - 1\big)\right] \qquad \text{and} \qquad \overline{W} = p_\pi(A|X) \cdot \left[1 + \Gamma\big(\widehat{p}(A|X)^{-1} - 1\big)\right]. \tag{12}$$

The model $\widehat{p}(A|X)$ is typically estimated from prior observable data. The above bounds are functions of $X$ and $A$ drawn from the training distribution (4). In order to ensure finite-sample guarantees, we randomly split the training data (1) into two separate sets,

$$\mathcal{D} = \mathcal{D}_0 \cup \mathcal{D}_1,$$

with samples of sizes $n_0$ and $n - n_0$, respectively. The set $\mathcal{D}_1$ is used to form the function

$$\widehat{F}(\ell; \mathcal{D}_1, w) = \frac{\sum_{i=n_0+1}^{n} \underline{W}_i \mathbb{1}(L_i \leq \ell)}{\sum_{i=n_0+1}^{n} \underline{W}_i \mathbb{1}(L_i \leq \ell) + \sum_{i=n_0+1}^{n} \overline{W}_i \mathbb{1}(L_i > \ell) + w}, \tag{13}$$

as a proxy for the unknown cdf of the out-of-sample loss $L_{n+1}$ and $w > 0$ represents the ratio in (11). We use the set $\mathcal{D}_0$ to construct an upper bound on this unknown ratio. As the following result shows, this enables us to construct a valid limit $\ell_\alpha$ on the future loss $L_{n+1}$.

**Theorem 4.1.** *Define the following quantile function of* (13) *as*

$$\ell_{\alpha,\beta} = \inf\left\{\ell : \widehat{F}(\ell; \mathcal{D}_1, \overline{w}_\beta(\mathcal{D}_0)) \geq \frac{1-\alpha}{1-\beta}\right\}, \tag{14}$$

*where*

$$\overline{w}_\beta(\mathcal{D}_0) = \begin{cases} \overline{W}_{[\lceil (n_0+1)(1-\beta)\rceil]}, & \lceil (n_0+1)(1-\beta)\rceil \leq n_0, \\ \infty, & otherwise, \end{cases} \tag{15}$$

*and* $\overline{W}_{[k]}$ *denotes the kth order statistic of* $(\overline{W}_i)_{i=1}^{n_0}$.

*For any miscoverage probability* $\alpha \in (0,1)$, *construct*

$$\boxed{\ell_\alpha(\mathcal{D}) = \min_{\beta:0<\beta<\alpha} \ell_{\alpha,\beta},} \tag{16}$$

*which a valid limit on the out-of-sample loss* $L_{n+1}$ *with a probability of at least* $1 - \alpha$ *as specified in* (2).

The proof of Theorem 4.1 can be found in Appendix A.1.

The variable $\overline{w}_\beta(\mathcal{D}_0)$ in (15) provides a statistically valid upper bound on the unknown ratio in (11) where $\beta$ only specifies its confidence level. For *any* given value of $0 < \beta < \alpha$, $\ell_{\alpha,\beta}$ in (14) bounds the loss $L_{n+1}$ with a coverage probability of at least $1 - \alpha$. Consequently, (2) chooses the tightest achievable limit.

**Remark 1.** *When* $\Gamma = 1$, *then* $\underline{W} = \overline{W}$ *in* (11). *Therefore* (13) *can be thought of as an inverse propensity weighted cdf, where the weights are normalized to sum to unity.*

We now turn to the implementation of (16). We note that (13) is a piecewise constant function in $\ell$ and that $\overline{w}_\beta$ in (15) is a piecewise constant function in $\beta$. Therefore (14) and subsequently (16) can be solved by evaluating functions along a discrete set of points.

The computation of a limit curve is summarized in Algorithm 1, using a discrete grid of miscoverage levels $\alpha$. Given a model $\widehat{p}(A|X)$ and a range of odds divergence bounds $\Gamma$, Algorithm 1 produces a set of corresponding limit curves.

## 5 Numerical experiments

In the experiments below, we evaluate policies using the limit curves $(\alpha, \ell_\alpha)$. We quantify how increasing the credibility of our model assumption, i.e., by increasing $\Gamma$, affects the informativeness of the limit curve using (7). We also consider the coverage probability of the curves:

$$\text{Miscoverage gap} = \alpha - \mathbb{P}_\pi\{L_{n+1} > \ell_\alpha(\mathcal{D})\}. \tag{17}$$

---

**Algorithm 1** Limit curve of policy $\pi$

---

**Input:** Policy $p_\pi(A|X)$, training data $\mathcal{D}$, model $\widehat{p}(A|X)$, bound $\Gamma \geq 1$ and sample split size $n_0$.
1: Randomly split $\mathcal{D}$ into $\mathcal{D}_0$ and $\mathcal{D}_1$.
2: **for** $\alpha \in \{0, \ldots, 1\}$ **do**
3:     **for** $\beta \in \{0, \ldots, \alpha\}$ **do**
4:         Compute $\overline{w}_\beta$ using (15).
5:         Compute $\ell_{\alpha,\beta}$ using (14).
6:     **end for**
7:     Set $\ell_\alpha$ to the smallest $\ell_{\alpha,\beta}$ above.
8:     Store $(\alpha, \ell_\alpha)$.
9: **end for**
**Output:** $\{(\alpha, \ell_\alpha)\}$

---

When this gap is positive, the limit is *conservative* and when the gap is negative the limit is *invalid*, respectively, at level $\alpha$.

A natural benchmark for the proposed limit (16) in this problem setting is the estimated quantile

$$\ell_\alpha(\mathcal{D}) = \inf\left\{\ell : \widehat{F}_{\text{IPW}}(\ell; \mathcal{D}) \geq 1 - \alpha\right\}, \tag{18}$$

using the inverse propensity weighted cdf-estimator (10).

In all examples below, the limit (16) is computed using sample splits of equal size, i.e., $n_0 = \lceil n/2 \rceil$. An additional experiment using data from the Infant Health and Development Program (IHDP) can be found in Appendix A.2.

The code used for the experiments is made available here `https://github.com/sofiaek/off-policy-evaluation`.

### 5.1 Synthetic data

In the first example, we consider synthetic data in order to evaluate the coverage of the derived limit curves. We use a simulation setting similar to Jin et al. (2023). The miscoverage gap (17) is estimated by Monte Carlo simulation using 1000 runs and for each run drawing independent 1000 new samples $(X_{n+1}, U_{n+1}, A_{n+1}, L_{n+1})$.

We consider a population of individuals with two-dimensional covariates distributed uniformly as

$$X = \begin{bmatrix} X_1 \\ X_2 \end{bmatrix} \sim \mathcal{U}(0,1)^2.$$

The actions are binary $A \in \{0, 1\}$ corresponding to 'not treat' and 'treat', respectively. We want to evaluate a deterministic target policy, described by

$$p_\pi(A = 0|X) = \mathbb{1}(X_1 X_2 \geq \tau), \tag{19}$$

for different $\tau \in [0, 1]$. That is, all individuals whose covariate product $X_1 X_2$ falls below $\tau$ are treated. Note that $\tau = 0$ corresponds a 'treat none' policy ($A \equiv 0$ for all $X$) and $\tau = 1$ corresponds to a 'treat all' policy ($A \equiv 1$ for all $X$). Below we discuss the resulting losses under this policy using observational data with sample sizes $n \in \{250, 500, 1000\}$.

**Case: Known past policy ($\Gamma = 1$).** In the first scenario, we assume that the past policy is known exactly and there is no unmeasured confounding.

For the training data, the past policy has selected actions as a Bernoulli process:

$$p(A = 0|X) \equiv \widehat{p}(A = 0|X) = f\left(c(X_1 X_2 + 1)\right), \quad c \in \left[\frac{1}{2}, 2\right], \tag{20}$$

where $f(\cdot)$ is the sigmoid function. The conditional loss distribution is given by

$$(L|A = 0, X) \sim \mathcal{N}(1 - X_1 X_2, 0.1) \quad \text{and} \quad (L|A = 1, X) \sim \mathcal{N}(X_1 X_2, 0.1),$$

and thus $L_{\max} = \infty$. We consider three configurations $c$ of past policies (20), which yield inverse propensity weights in three ranges: $\frac{1}{p_1(A|X)} < 3.72$ $(c = 1/2)$, $\frac{1}{p_2(A|X)} < 8.39$ $(c = 1)$, and $\frac{1}{p_3(A|X)} < 55.6$ $(c = 2)$. Thus we anticipate $p_3(A|X)$ to be the most challenging case.

Here we evaluate three target policies $\tau = \{0, 0.5, 1\}$ in (19) and present their resulting limit curves using data from different past policies (20), see Figure 3. The dashed line shows the corresponding past policy. The limit curves for a given target policy are quite similar across training distributions and are still informative at the 90% level using (7). The main difference is in the inferred tail losses and is notable for when $\tau = 1$ under the more challenging past policy $p_3(A|X)$.

We now turn to the evaluating miscoverage gap (17). Figure 4 presents the gaps for target policy $\tau = 0.5$ in (19). The solid lines illustrate the proposed method and the dashed lines show the benchmark (18). We see that the proposed method is slightly conservative, but remains valid for all $\alpha$. In contrast, the benchmark is not valid in the tail of the distribution, but is less conservative for higher $\alpha$ in this well-specified case.

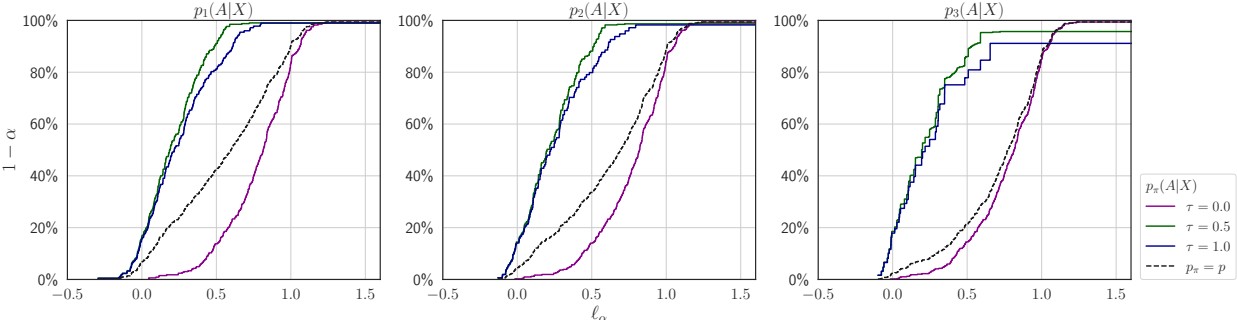

Figure 3: Limit curves when the past policy is known ($\Gamma = 1$) for three different potential target policies (i.e. $\tau = \{0.0, 0.5, 1.0\}$ in (19)). Dashed curve denotes the past policy. $n = 1000$.

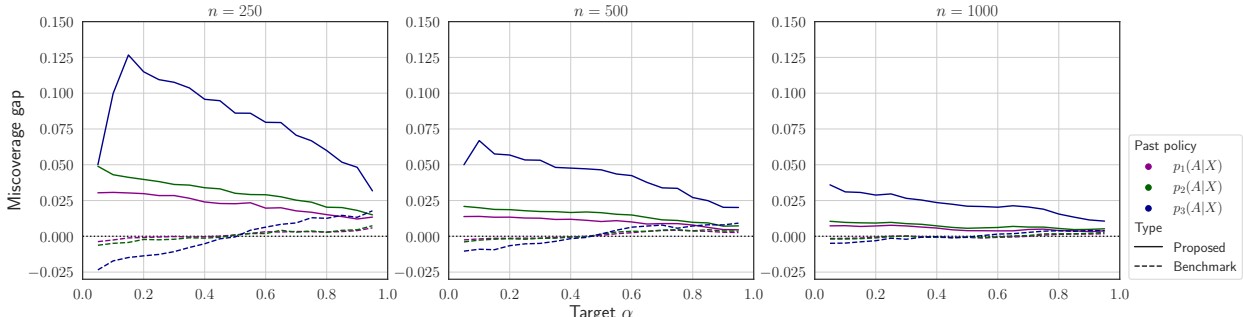

Figure 4: Miscoverage gaps (17) versus $\alpha$, when the past policy is known ($\Gamma = 1$). Dashed curve denotes the benchmark (18).

**Case: Estimated past policy ($\Gamma > 1$).** In the second scenario, we assume that we only have access to an estimate $\widehat{p}(A|X)$ (given by (20)) and that there is unmeasured confounding in the training distribution. To render visually distinct curves from the previous case, we consider here a rather extreme case of confounding following Jin et al. (2023).

Specifically, we have an unobserved variable drawn as

$$(U|X) \sim \mathcal{N}(0, 0.1(X_1 + X_2)),$$

and the loss $(L|A, X, U)$ is generated as

$$L = \begin{cases} 1 - X_1 X_2 + U, & A = 0, \\ X_1 X_2 + U, & A = 1. \end{cases}$$

We define the past policy in a manner that enables us to control the divergence from the nominal model $\widehat{p}(A|X)$ in (20):

$$\begin{aligned} p(A = 0|X, U) = & \mathbb{1}(U \leq t(X)) \left[ 1 + \Gamma_0^{-1}\big(\widehat{p}(A = 0|X)^{-1} - 1\big) \right] \\ & + \mathbb{1}(U > t(X)) \left[ 1 + \Gamma_0\big(\widehat{p}(A = 0|X)^{-1} - 1\big) \right], \end{aligned} \qquad (21)$$

where the threshold function $t(X)$ is designed empirically to ensure that the resulting median loss of the past policy for $A = 1$ is maximized. Our design of the past policy can be seen as the worst case among all unknown past policies that diverge by a factor $\Gamma_0$ in (6). We fix $\Gamma_0 = 2$ here, but treat it as unknown.

For simplicity, we consider a 'treat all' target policy ($\tau = 1$). Its limit curves, under different assumed odds divergence bounds $\Gamma = \{1, 2, 3\}$, are presented in Figure 5. Note that in the case of unmeasured confounding, the resulting curves differ notably across the training distributions unlike in Figure 3. We see that for the first and second distributions, the informativeness of all curves stays around the 90% level. However, in the more extreme third case, the informativeness drops to just above the 60% level when we increase the credibility of our model assumption to an odds bound of $\Gamma = 3$. This example illustrates an inherent trade-off between credibility and informativeness.

Figure 6 validates our guarantees using data drawn from $p_1(A|X)$. When $\Gamma \geq \Gamma_0 = 2$, the limit curves are valid and as $\Gamma$ increases to 3, the limits become quite conservative. Note that the conservativeness persists even as the sample size $n$ is increased fourfold. For $\Gamma = 1$, there is no guarantee of coverage and in this case the limit curve is invalid. The benchmark does not take the unmeasured confounding into account and is consequently invalid throughout.

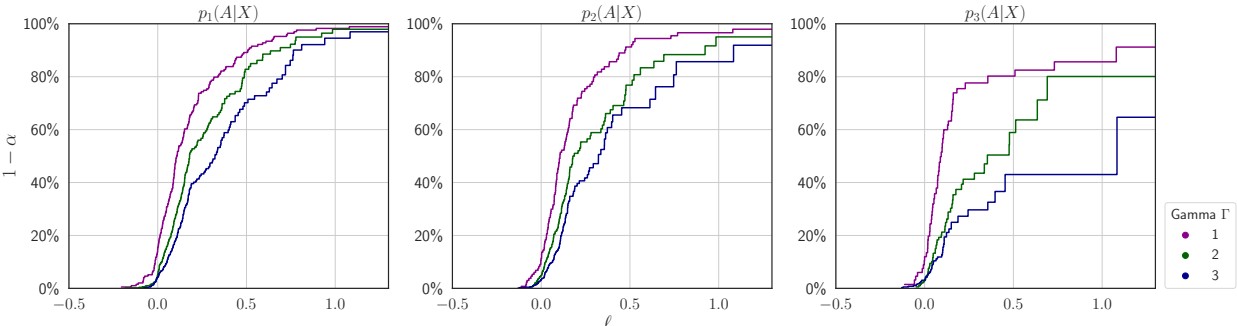

Figure 5: Limit curves for 'treat all' target policy using odds bounds $\Gamma = \{1, 2, 3\}$, when the past policy is unknown and subject to unmeasured confounding ($\Gamma_0 = 2$ in (21)). $n = 1000$.

## 5.2 Real data

In the second example, we use data from the National Health and Nutrition Examination Survey (NHANES) for the years 2013-2014 to illustrate the use of the proposed method. Following Zhao et al. (2019), we study the effect of seafood consumption on blood mercury levels. The action $A$ indicates whether a person has a low or high consumption of fish or shellfish ($\leq 1$ vs. $> 12$ servings in the past month) and the loss $L$ is the total concentration of blood mercury (in $\mu$g/L). There are 8 covariates in $X$ for each person: gender, age, income, whether income is missing or not, race, education, ever smoked and number of cigarettes smoked last month. 1 individual with missing education data and 7 with missing smoking data are omitted. 175 individuals have missing income data, for them we impute the median income. This preprocessing results

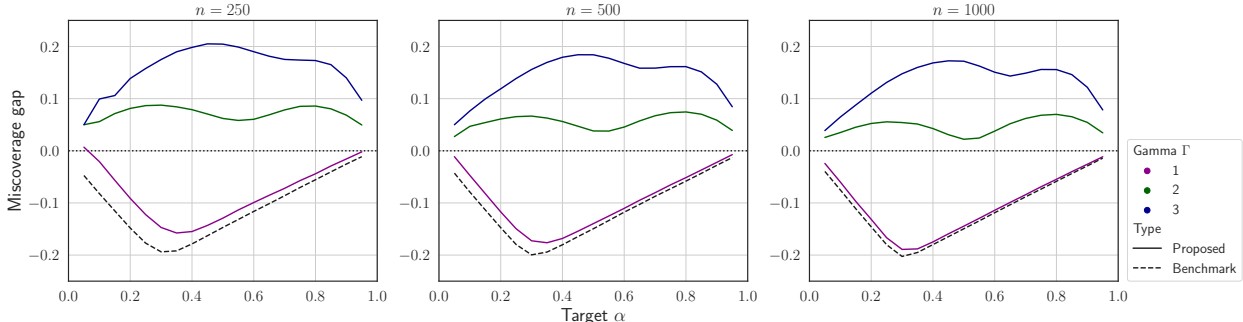

Figure 6: Miscoverage gaps (17) versus $\alpha$, when the past policy is unknown and subject to unmeasured confounding. Dashed curve denotes the benchmark (18) which does not take confounding into account. Note that $\Gamma = 1$ assumes there is no odds divergence for our model, which in this case leads to an invalid limit curve.

in a data set of 1107 individuals of which 21% follow a high fish consumption diet and 79% follow a low consumption diet, see Zhao et al. (2018). We use a fitted logistic regression model for $\hat{p}(A|X)$, where the continuous covariates in $X$ have been standardized.

We compare two policies, $\pi_0$ and $\pi_1$, corresponding to low ($A \equiv 0$) and high ($A \equiv 1$) seafood consumption, respectively. Their corresponding limit curves are presented in Figure 7 under different odds divergences bounds $\Gamma$ on the model $\hat{p}(A|X)$. In the most extreme case, we consider the nominal odds diverging by at most a factor $\Gamma = 3$. For reference, we display the limit curve under the unknown consumption policy in the population. We see that lower mercury levels can be certified by low seafood consumption. Note that a guidance value for mercury levels is 8 $\mu$g/L for women of child-bearing age and 20 $\mu$g/L for women $\geq 50$ years (Kales & Thompson, 2016). If we were to evaluate policies based on the expected loss alone, then high seafood consumption for women has an expected loss of approximately 3.3 $\mu$g/L and could be deemed safe. However, this figure masks the tail losses that substantial proportion of women may face under such a policy. For a high consumption policy, our method could only certify that approximately 80% ($\Gamma = 1$) to 50% ($\Gamma = 3$) of females would stay below the guideline level of 8 $\mu$g/L. In contrast, for the low consumption policy, the corresponding figure is approximately 95% (for $1 \leq \Gamma \leq 3$).

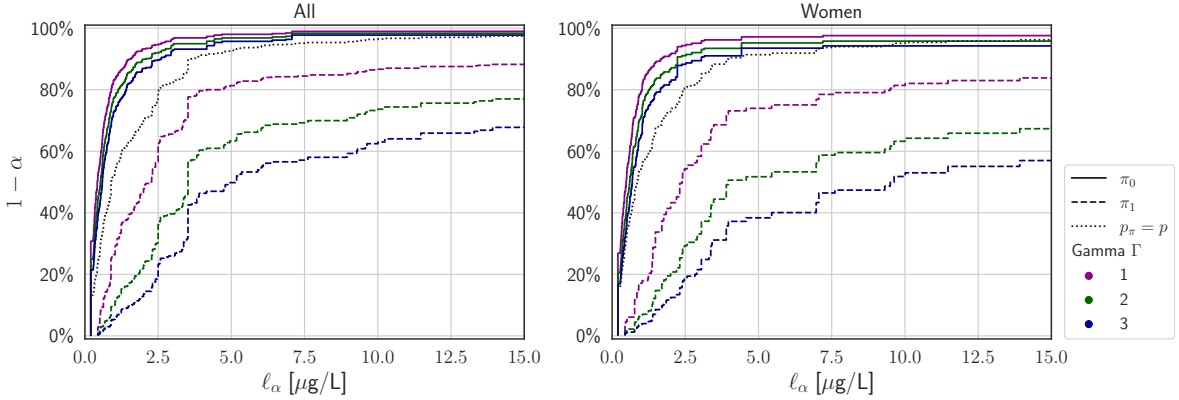

Figure 7: Limit curves for blood mercury levels in a population under different seafood consumption policies. Left: Overall population. Right: Female population. We compare two policies: low ($\pi_0$) and high ($\pi_1$) consumption. The credibility of the assumptions increases with the odds bound $\Gamma = \{1, 2, 3\}$. The past policy is indicated by dotted curves.

# 6 Conclusion

We have considered the problem of off-policy evaluation, i.e., making valid inferences about a target policy using past observational data obtained under a different decision process with a possibly unknown policy. Using the marginal sensitivity model, we derived a sample-splitting method that provides limit curves with finite-sample coverage guarantees even under model misspecifications, including unmeasured confounding. The validity, informativeness, and conservativeness of the resulting limit curves were demonstrated in the numerical experiments.

**Acknowledgments**

We thank the anonymous reviewers from TMLR for their constructive feedback. This research was partially supported by the Wallenberg AI, Autonomous Systems and Software Program (WASP) funded by Knut and Alice Wallenberg Foundation, and the Swedish Research Council under contract 2021-05022.

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

# A  Appendix

## A.1  Proof of Theorem 4.1

The first part of the proof builds on techniques used to derive weighted conformal prediction intervals in Tibshirani et al. (2019).

Let us consider a sequence of $n - n_0$ samples drawn from (4) followed by a new sample drawn from (3), i.e.,

$$\mathcal{D}_+ = \big((X_{n_0+1}, U_{n_0+1}, A_{n_0+1}, L_{n_0+1}), \ldots, (X_n, U_n, A_n, L_n), (X_{n+1}, U_{n+1}, A_{n+1}, L_{n+1})\big).$$

The joint distribution of this sequence can be expressed using:

$$\prod_{i=n_+}^{n} p(x_i, u_i, a_i, \ell_i) \cdot p(x_{n+1}, u_{n+1}, a_{n+1}, \ell_{n+1}) w_{n+1} = p(\mathcal{D}_+) w_{n+1} = p(\mathcal{S}_+) w_{n+1}, \tag{22}$$

where $n_+ = n_0 + 1$ for notational simplicity, $\mathcal{S}_+$ is an unordered set of elements from $\mathcal{D}_+$, and the weight

$$w_i = \frac{p_\pi(x_i, u_i, a_i, \ell_i)}{p(x_i, u_i, a_i, \ell_i)},$$

is the (unobservable) ratio (5) that quantifies the distribution shift from training to target distribution. We shall use the expression for the joint distribution to derive the distribution function for the new loss $L_{n+1}$.

Suppose we are given the unordered set $\mathcal{S}_+$ alone, then the particular sequence $\mathcal{D}_+$ is unknown. Let $E_i$ denote the event that the sample $(X_{n+1}, U_{n+1}, A_{n+1}, L_{n+1})$ equals the $i$th sample $(x_i, u_i, a_i, \ell_i)$ in the unknown sequence $\mathcal{D}_+$. We consider all possible sequences $\mathcal{D}_+$ obtained by permutations $\sigma$ of elements in the set $\mathcal{S}_+$. Using the joint distribution in (22), the joint probability of the event $E_i$ and $\mathcal{S}_+$ is then

$$\mathbb{P}\{E_i, \mathcal{S}_+\} = \sum_{\sigma : \sigma(n+1) = n+i} p(\mathcal{S}_+) w_i = p(\mathcal{S}_+) w_i n!,$$

where we have used the symbol $\mathbb{P}$ for the samples that are drawn from two different processes (4) and (3).

The conditional probability of event $E_i$ can now be expressed as

$$p_i = \mathbb{P}\{E_i | \mathcal{S}_+\} = \frac{\mathbb{P}\{E_i, \mathcal{S}_+\}}{\sum_{j=n_+}^{n+1} \mathbb{P}\{E_j, \mathcal{S}_+\}} = \frac{w_i}{\sum_{j=n_+}^{n+1} w_j},$$

where the first equality follows from the law of total probability. The probability that the loss $L_{n+1}$ of the new sample equals $\ell_i$, when conditioning on the unordered set $\mathcal{S}_+$, is equal to

$$\mathbb{P}\{L_{n+1} = \ell_i | \mathcal{S}_+\} = \mathbb{P}\{E_i | \mathcal{S}_+\} = p_i.$$

Thus conditional on $\mathcal{S}_+$, the new loss $L_{n+1}$ has the following cdf:

$$\mathbb{P}\{L_{n+1} \leq \ell | \mathcal{S}_+\} = \sum_{i=n_+}^{n+1} p_i \mathbb{1}(\ell_i \leq \ell) = \frac{\sum_{i=n_+}^{n+1} w_i \mathbb{1}(L_i \leq \ell)}{\sum_{i=n_+}^{n+1} w_i}. \tag{23}$$

Before marginalizing out $\mathcal{S}_+$ from (23), we consider a limit $\ell$ that is a function of the observable elements in $\mathcal{S}_+$. For this part, we will build on the proof technique in (Jin et al., 2023, thm. 2.2).

Specifically, using (13) we define the following limit:

$$\ell_\alpha(\mathcal{D}_1, \overline{W}_{n+1}) = \inf\left\{\ell : \widehat{F}(\ell; \mathcal{D}_1, \overline{W}_{n+1}) \geq \frac{1-\alpha}{1-\beta}\right\}, \tag{24}$$

for any $0 < \beta < \alpha$, where $\overline{W}_{n+1} \geq W_{n+1}$ is given in (12). Now insert the limit $\ell_\alpha(\mathcal{D}_1, \overline{W}_{n+1})$ into (23) and use the law of total expectation to marginalize out $\mathcal{S}_+$:

$$\mathbb{P}\{L_{n+1} \leq \ell_\alpha(\mathcal{D}_1, \overline{W}_{n+1})\} = \mathbb{E}\left[\mathbb{P}\{L_{n+1} \leq \ell_\alpha(\mathcal{D}_1, \overline{W}_{n+1}) | \mathcal{S}_+\}\right]$$

$$= \mathbb{E}\left[\frac{\sum_{i=n_+}^{n+1} W_i \mathbb{1}(L_i \leq \ell_\alpha(\mathcal{D}_1, \overline{W}_{n+1}))}{\sum_{i=n_+}^{n+1} W_i}\right].$$

We now proceed to lower bound this probability. Note that by construction:

$$\mathbb{E}\left[\widehat{F}(\ell_\alpha; \mathcal{D}_1, \overline{W}_{n+1})\right] = \mathbb{E}\left[\frac{\sum_{i \in \mathcal{D}_1} \underline{W}_i \mathbb{1}(L_i \leq \ell_\alpha)}{\sum_{i \in \mathcal{D}_1} \underline{W}_i \mathbb{1}(L_i \leq \ell_\alpha) + \sum_{i \in \mathcal{D}_1} \overline{W}_i \mathbb{1}(L_i > \ell_\alpha) + \overline{W}_{n+1}}\right] \geq \frac{(1-\alpha)}{(1-\beta)}.$$

Using this expression, we have that

$$\mathbb{P}\{L_{n+1} \leq \ell_\alpha(\mathcal{D}_1, \overline{W}_{n+1})\} - \frac{(1-\alpha)}{(1-\beta)}$$

$$\geq \mathbb{E}\left[\frac{\sum_{i=n_+}^{n+1} W_i \mathbb{1}(L_i \leq \ell_\alpha)}{\sum_{i=n_+}^{n+1} W_i}\right] - \mathbb{E}\left[\frac{\sum_{i=n_+}^{n} \underline{W}_i \mathbb{1}(L_i \leq \ell_\alpha)}{\sum_{i=n_+}^{n} \underline{W}_i \mathbb{1}(L_i \leq \ell_\alpha) + \sum_{i=n_+}^{n} \overline{W}_i \mathbb{1}(L_i > \ell_\alpha) + \overline{W}_{n+1}}\right]$$

$$= \mathbb{E}\left[\frac{(*)}{\left[\sum_{i=n_+}^{n+1} W_i\right]\left[\sum_{i=n_+}^{n} \underline{W}_i \mathbb{1}(L_i \leq \ell_\alpha) + \sum_{i=n_+}^{n} \overline{W}_i \mathbb{1}(L_i > \ell_\alpha) + \overline{W}_{n+1}\right]}\right],$$

where

$$(*) = \left[\sum_{i=n_+}^{n+1} W_i \mathbb{1}(L_i \leq \ell_\alpha)\right]\left[\sum_{i=n_+}^{n} \underline{W}_i \mathbb{1}(L_i \leq \ell_\alpha) + \sum_{i=n_+}^{n} \overline{W}_i \mathbb{1}(L_i > \ell_\alpha) + \overline{W}_{n+1}\right] - \left[\sum_{i=n_+}^{n} \underline{W}_i \mathbb{1}(L_i \leq \ell_\alpha)\right]\left[\sum_{i=n_+}^{n+1} W_i\right]$$

$$\geq \left[\sum_{i=n_+}^{n} W_i \mathbb{1}(L_i \leq \ell_\alpha)\right]\left[\sum_{i=n_+}^{n} \underline{W}_i \mathbb{1}(L_i \leq \ell_\alpha) + \sum_{i=n_+}^{n} \overline{W}_i \mathbb{1}(L_i > \ell_\alpha) + \overline{W}_{n+1}\right] - \left[\sum_{i=n_+}^{n} \underline{W}_i \mathbb{1}(L_i \leq \ell_\alpha)\right]\left[\sum_{i=n_+}^{n+1} W_i\right]$$

$$\geq \left[\sum_{i=n_+}^{n} W_i \mathbb{1}(L_i \leq \ell_\alpha)\right]\left[\sum_{i=n_+}^{n} \overline{W}_i \mathbb{1}(L_i > \ell_\alpha) + \overline{W}_{n+1}\right] - \left[\sum_{i=n_+}^{n} \underline{W}_i \mathbb{1}(L_i \leq \ell_\alpha)\right]\left[\sum_{i=n_+}^{n} W_i \mathbb{1}(L_i > \ell_\alpha) + W_{n+1}\right]$$

$$\geq \left[\sum_{i=n_+}^{n} W_i \mathbb{1}(L_i \leq \ell_\alpha)\right]\left[\sum_{i=n_+}^{n} W_i \mathbb{1}(L_i > \ell_\alpha) + W_{n+1}\right] - \left[\sum_{i=n_+}^{n} W_i \mathbb{1}(L_i \leq \ell_\alpha)\right]\left[\sum_{i=n_+}^{n} W_i \mathbb{1}(L_i > \ell_\alpha) + W_{n+1}\right]$$

$$= 0,$$

using the bounds in (11) to get the third inequality. Therefore we obtain a valid limit:

$$\mathbb{P}\{L_{n+1} \leq \ell_\alpha(\mathcal{D}_1, \overline{W}_{n+1})\} \geq \frac{(1-\alpha)}{(1-\beta)}. \tag{25}$$

However, $\overline{W}_{n+1}$ depends on a new sample drawn from the training distribution and thus the limit is unusable. In lieu of $\overline{W}_{n+1}$, we use $\overline{w}_\beta(\mathcal{D}_0)$ in (15) to define the modified limit

$$\ell_\alpha(\mathcal{D}) = \inf\left\{\ell : \widehat{F}(\ell; \mathcal{D}_1, \overline{w}_\beta(\mathcal{D}_0)) \geq \frac{1-\alpha}{1-\beta}\right\}. \tag{26}$$

Comparing it with (24), we see that

$$\ell_\alpha(\mathcal{D}) \geq \ell_\alpha(\mathcal{D}_1, \overline{W}_{n+1}), \tag{27}$$

whenever $\overline{W}_{n+1} \leq \overline{w}_\beta(\mathcal{D}_0)$. By the construction in (15), the probability of this event is lower bounded by

$$\mathbb{P}\{\overline{W}_{n+1} \leq \overline{w}_\beta(\mathcal{D}_0)\} \geq 1 - \beta, \tag{28}$$

see Vovk et al. (2005); Lei et al. (2018).

We use this property to lower bound the probability of $L_{n+1} \leq \ell_\alpha(\mathcal{D})$. First, note that

$$\mathbb{P}\{L_{n+1} \leq \ell_\alpha(\mathcal{D})\} = \mathbb{P}\{L_{n+1} \leq \ell_\alpha(\mathcal{D}) \mid \overline{W}_{n+1} \leq \overline{w}^\beta(\mathcal{D}_0)\}\,\mathbb{P}\{\overline{W}_{n+1} \leq \overline{w}^\beta(\mathcal{D}_0)\}$$
$$+ \mathbb{P}\{L_{n+1} \leq \ell_\alpha(\mathcal{D}) \mid \overline{W}_{n+1} > \overline{w}^\beta(\mathcal{D}_0)\}\,\mathbb{P}\{\overline{W}_{n+1} > \overline{w}^\beta(\mathcal{D}_0)\}$$
$$\geq \mathbb{P}\{L_{n+1} \leq \ell_\alpha(\mathcal{D}) \mid \overline{W}_{n+1} \leq \overline{w}^\beta(\mathcal{D}_0)\}\,\mathbb{P}\{\overline{W}_{n+1} \leq \overline{w}^\beta(\mathcal{D}_0)\} + 0.$$

The first factor can be lower bounded using (27), so that

$$
\begin{aligned}
\mathbb{P}\{L_{n+1} \leq \ell_\alpha(\mathcal{D})\} &\geq \mathbb{P}\{L_{n+1} \leq \ell_\alpha(\mathcal{D}_1, \overline{W}_{n+1}) \mid \overline{W}_{n+1} \leq \overline{w}^\beta(\mathcal{D}_0)\} \, \mathbb{P}\{\overline{W}_{n+1} \leq \overline{w}^\beta(\mathcal{D}_0)\} \\
&= \mathbb{P}\{L_{n+1} \leq \ell_\alpha(\mathcal{D}_1, \overline{W}_{n+1})\} \, \mathbb{P}\{\overline{W}_{n+1} \leq \overline{w}^\beta(\mathcal{D}_0)\} \\
&\geq \frac{(1-\alpha)}{(1-\beta)} \, \mathbb{P}\{\overline{W}_{n+1} \leq \overline{w}^\beta(\mathcal{D}_0)\} \\
&\geq 1 - \alpha.
\end{aligned}
\tag{29}
$$

The second line follows from using sample splitting, which ensures that $L_{n+1} \leq \ell_\alpha(\mathcal{D}_1, \overline{W}_{n+1})$ and $\overline{W}_{n+1} \leq \overline{w}^\beta(\mathcal{D}_0)$ are independent events. The third and fourth lines follow from (25) and (28), respectively. Since (29) holds for any $0 < \beta < \alpha$, we choose $\beta$ in (26) that yields the tightest limit, cf. (16). Note that $\mathbb{P}$ here represents the probability over samples drawn from both (4) and (3). Since $L_{n+1}$ is drawn from (3), we can write (29) as in (2) for notational clarity.

## A.2 Semi-real data

The Infant Health and Development Program (IHDP) investigated the impact of early childhood interventions on the health of low birth-weight and premature infants (Health & Program, 1990). Hill (2011) used this study to assemble a data set of 25 covariates $X$ that measured various aspects of the children and their mothers such as birth weight, weeks born preterm, head circumference and age of mother, etc. These covariates were standardized to have a zero mean and unit standard deviation. The data set also includes whether each child received special medical care or not, denoted by action $A$. The original study was a randomised control study, but the data were imbalanced as a non-random subset of the treated population was removed. This unbalanced data set contains 747 children, including 139 treated and 608 control units.

The outcome was the child's cognitive development score, which we reverse it to an underdevelopment score to treat it as a loss $L$. The outcome is generated synthetically and we use 'response surface A' from Hill (2011):

$$
L|A=0, X \sim \mathcal{N}(-\varphi^\top X, 1), \quad L|A=1, X \sim \mathcal{N}(-(\varphi^\top X + 4), 1).
$$

The vector $\varphi$ is discrete valued with elements drawn from $\{0, 1, 2, 3, 4\}$ with probabilities $\{0.5, 0.2, 0.15, 0.1, 0.05\}$. We use a fitted logistic regression model for $\hat{p}(A|X)$. The limit (16) is computed using sample splits where $n_0 = 0.1n$. Another $0.1n$ samples of the data is randomly used to evaluate the policies in Figure 9.

We compare two policies: the child did not receive special medical care $\pi_0$ ('treat none') or the child received special medical care $\pi_1$ ('treat all'). The limit curves are presented in Figure 8 under different odds divergence bounds $\Gamma = \{1, 1.5, 2\}$ on the nominal model. When $\Gamma = 1$, the 'treat all' policy is the policy resulting in the lowest limit curve. When $\Gamma$ is increased to 2 the limit curves are closer to each other, but the 'treat none' policy is instead the policy with the lowest limit curve.

Figure 9 validates our guarantee in (2). The miscoverage gap (17) is estimated using 1000 simulation runs where we draw a new vector $\varphi$ and a new underdevelopment score $L$ for each run. The proposed method is valid for all values of $\Gamma$, but is more conservative for the higher values. The benchmark does not take the unmeasured confounding into account and is invalid for the 'treat all' policy (right).

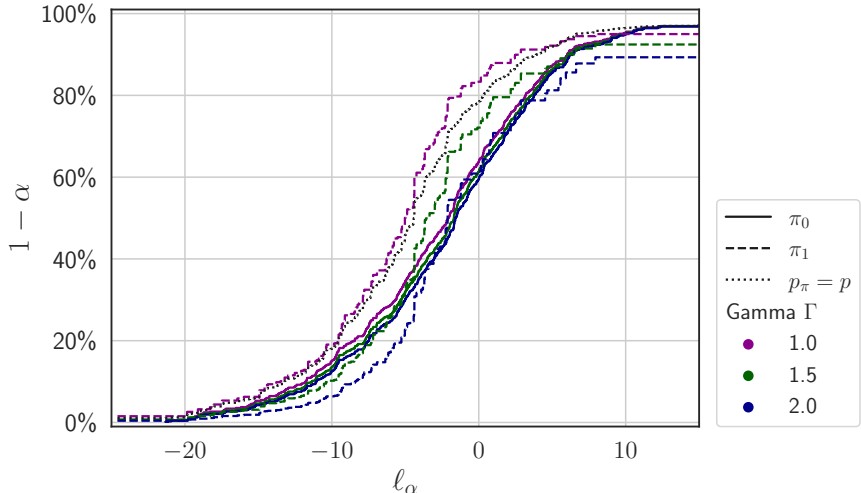

Figure 8: Limit curves for a synthetic underdevelopment score in a population under different policies. We compare two policies: the child did not receive special medical care $\pi_0$ or the child received special medical care $\pi_1$. The past policy is indicated by dotted curves ($p_\pi = p$). The credibility of the assumptions increases with the odds bound $\Gamma = \{1, 1.5, 2\}$.

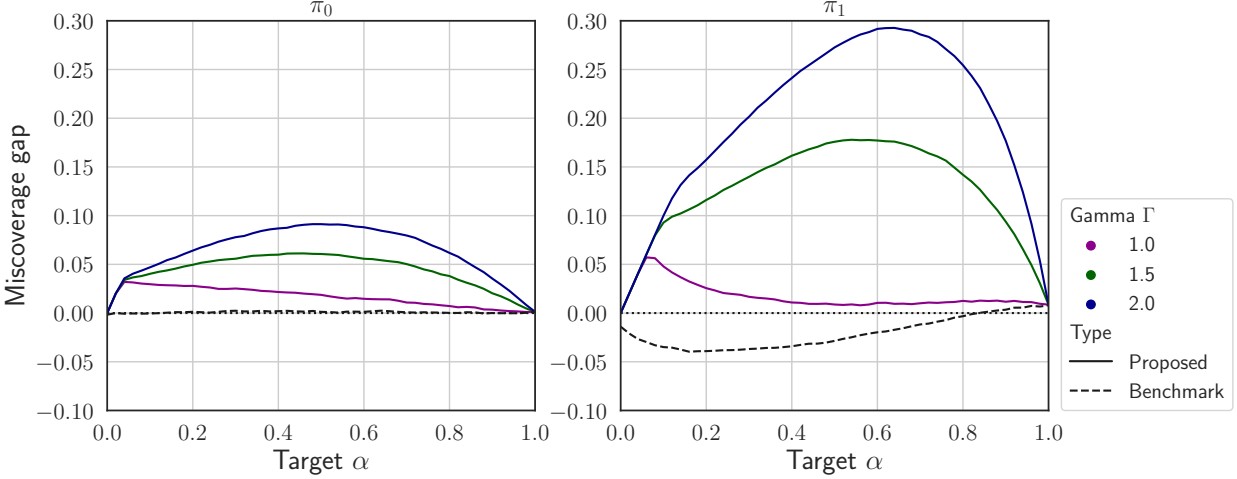

Figure 9: Miscoverage gaps (17) versus $\alpha$ for data from the Infant Health and Development Program (IHDP) under two different policies: the child did not receive special medical care $\pi_0$ or the child received special medical care $\pi_1$. Dashed curve denotes the benchmark (18) which does not take confounding or modelling errors into account. The credibility of the assumptions increases with the odds bound $\Gamma = \{1, 1.5, 2\}$.

