# OpenReview forum: "Off-Policy Evaluation with Out-of-Sample Guarantees"
_TMLR — Accepted by TMLR_

### Review · Reviewer_ggu1 · 2023-03-29

**Summary Of Contributions:**

The paper looks at the problem of off-policy evaluation and inference, where the observed data might be subject to unobserved confounding. The key assumption being made to get around this issue of unobserved confounding is to assume a known bound on the odds ratio (Eqn 6). Under this assumption, the authors theoretically show how techniques from conformal inference can be used to get coverage over the entire distribution of loss/outcomes. An empirical investigation of a toy and a real-world example is provided to support the claim.

**Audience:**

Yes

**Broader Impact Concerns:**

Since this paper deals with topics that aim at safety and credibility, it might benefit from more discussion on assumption implied by Eqn 6 on the bounds of the odds ratio. How should this be chosen in practice, etc.?

**Claims And Evidence:**

Yes

**Requested Changes:**

1. Worth double check the use of phrase `external validity'. I have mostly seen the causal inference community uses that term for distribution mismatch of the covariates (distribution of X) or outcome (reward/cost/loss) models but the treatment policy is held constant. The setting being considered in the paper is where distribution of covariates and reward function is the same, but the treatment policy is different.

2. " loss no greater than ℓα(D) under policy π [with high confidence]."

3. Eqn 6 needs to made more precise. Is the ratio the worst case ratio over all possible X-U-A triplet?

4. Other related work around eqn 6 in the OPE literature (although for the mean, and not for quantile/CDF):

A Namkoong, Hongseok, et al. "Off-policy policy evaluation for sequential decisions under unobserved confounding." Advances in Neural Information Processing Systems

B Giguere, Stephen, et al. "Fairness guarantees under demographic shift." International Conference on Learning Representations. 2022.

5. Is the "nominal model" $\hat p(A|X)$ the probability of action given covariates estimated from the data, or is it the true value $\int_u p(A|X,u) du$? Interpretation of bounds in Eqn 6 will be different based on this, and this should be properly clarified.

6. I think Eqn 2 and problem statement needs a more nuanced discussion. If I understand it right, the citations for Chandak et al. and Huang et al. provide PAC-like (1-\epsilon accurate with 1-\delta confidence) coverage, i.e.,  $Pr_D(Pr_X(X < fn(D)) > 1- \epsilon) > 1-\delta$. The coverage in Eqn 2 is slightly different (and a weaker guarantee) as both $L_{k+1}$ and $D$ are random variables.   This is not to say that Eqn 2 is bad, but needs to be properly justified to the reader for why it is meaningful towards the motivation that preceded it.

See discussion around Eqn 5 (Vovk 2012), Eqn 1 and 2 (Danijel et al. 2020), and others (Sangdon et al.)

Vovk, Vladimir. "Conditional validity of inductive conformal predictors." Asian conference on machine learning. PMLR,

Kivaranovic, Danijel, Kory D. Johnson, and Hannes Leeb. "Adaptive, distribution-free prediction intervals for deep networks." International Conference on Artificial Intelligence and Statistics.

Park, Sangdon, et al. "PAC prediction sets under covariate shift." arXiv preprint arXiv:2106.09848 (2021).

7. While $\bar w$ is being estimated through data splitting, wouldn't the optimization process is Eqn 15 make the choice of $\bar w$ indirectly depend on both the partitions of the data? What is the intuition for introducing $\beta$ and what is the trade-off that $beta$ is balancing, e.g., why is $\beta=\alpha$ not adequate?

8. An important factor that separates this work from prior methods (Chandak et al. (2021) and Huang et al. (20221) is that it can deal with settings where the observational data has unobseved confounding, and this is possible by controlling the odds ratio (Eqn 6).

 Is there any particular reason to consider the ratio of the odds in Eqn 6 instead of bounding $p_\pi(A|X)/p(A|X,U)$ directly? Empirically, it _seems_ like a practitioner will have a better idea of the latter. Theoretically, could one not use these bounds in the works by Chandak et al. (2021) and Huang et al. (20221) to directly extend their bounds for the desired setting with unobserved confounders and unknown behavior policy? It feels like this would just affect the leading constant on the Hoeffding/DKW-like bound which currently depends on the max of the IPW/importance ratio's values.

9. I only got a chance to skim through the proof but I think as a reader it would be helpful if the authors can discuss intuitively what Eqn 13 and 15 are capturing as well. Here it will also be beneficial to rediscuss the setting where $\tau=1$, similar to how it was done in Section 3. These might make the draft more readable and perhaps the proof can be deferred to the appendix.

10. I am intrigued by the non-monotonous nature of the curves in Figure 4 and 6. Can authors share insight on why it is not monotonous?

11. Can authors discuss how to leverage their proposed method when one only requires to get coverage over the expected value of the loss/reward for the evaluation policy? (I suspect this might not be possible and would connect back to point 6 discussed earlier. But maybe I am wrong, and perhaps authors can correct me over here. )


**Strengths And Weaknesses:**

S1. An important problem that might be of high relevance to the audience

S2. Clearly written and to the point. Although I would have liked more discussions on assumptions, intuitions about the formulation, and clarifications (as mentioned below)

W1. I feel the motivation and the problem statement might be a bit disjoint.

W2. Technical novelty is limited.

---

> ### Author Response · Authors · 2023-05-02
> **Reply to Reviewer ggu1, part a)**
>
> We thank the reviewer for the constructive feedback and helpful comments.
>
> >\#1: “Worth double check the use of phrase `external validity'. I have mostly seen the causal inference community uses that term for distribution mismatch of the covariates (distribution of X) or outcome (reward/cost/loss) models but the treatment policy is held constant. The setting being considered in the paper is where distribution of covariates and reward function is the same, but the treatment policy is different.”
>
> To clarify, in the causal inference literature, ‘external validity’ refers to the problem of extrapolating from one population to another. Thus our usage is in agreement with that of the reviewer, see also our references to the textbooks by Westreich (2019) and Manski (2019). External validity is mainly a problem when using experimental/trial data. As we explain in the introduction, the main alternative to experimental trials is off-policy evaluation, which can avoid this problem by using observational data from a past decision process. Here the main problem is instead that of ‘internal validity’ – which is the focus of the paper.
>
> >\#2: “loss no greater than $\ell_{\alpha}(D)$ under policy $\pi$ [with high confidence].”
>
> We will revise the sentence accordingly.
>
> >\#3: “Eqn 6 needs to made more precise. Is the ratio the worst case ratio over all possible X-U-A triplet?”
>
> Yes, this is correct, (6) holds for all $(X,U,A)$ at any specified $\Gamma$. We will clarify this in the revised manuscript.
>
> >\#4: ”Other related work around eqn 6 in the OPE literature (although for the mean, and not for quantile/CDF): \
> -A Namkoong, Hongseok, et al. "Off-policy policy evaluation for sequential decisions under unobserved confounding." Advances in Neural Information Processing Systems \
> -B Giguere, Stephen, et al. "Fairness guarantees under demographic shift." International Conference on Learning Representations. 2022.”
>
> We thank the reviewer for finding these additional references, in particular the paper by Namkoong et al. (2020) is relevant since it uses a similar misspecification bound as in eq. (6). The paper by Giguere et al. (2022), while interesting, considers evaluating the fairness of predictive models under a specific type of distributional shifts which differs substantially from the off-policy evaluation problem.
>
> >\#5: ”Is the "nominal model" $\hat{p}(A|X)$ the probability of action given covariates estimated from the data, or is it the true value $\int_{u}p(A|X,u)du$? Interpretation of bounds in Eqn 6 will be different based on this, and this should be properly clarified.”
>
> Yes, the nominal model, $\hat{p}(A|X)$, is (typically) estimated from past data, as is stated in the paragraph following equation (5). For sake of clarity, we will repeat this point again in Section 4 when we describe the method.
>
> >\#6: ”I think Eqn 2 and problem statement needs a more nuanced discussion. If I understand it right, the citations for Chandak et al. and Huang et al. provide PAC-like (1-\epsilon accurate with 1-\delta confidence) coverage, i.e., $Pr_{D}(Pr_{X}(X \< fn(D)) \> 1 - \epsilon) \> 1 - \delta$. The coverage in Eqn 2 is slightly different (and a weaker guarantee) as both $L_{k+1}$ and $D$ are random variables. This is not to say that Eqn 2 is bad, but needs to be properly justified to the reader for why it is meaningful towards the motivation that preceded it.”
>
> Please note that neither the papers of Chandak et al. nor Huang et al. consider predictive limits on $L_{n+1}$. Rather, they provide confidence intervals for functionals $\rho(F)$ of a CDF $F$ (assuming well-specified models only), which are fundamentally different than predictive limit.
>
> The guarantee we consider in the paper, that is $P_{\pi} \\{ L_{n+1} \leq \ell_{\alpha}(\mathcal{D}) \\} \geq 1 - \alpha$ in (2), provides a statistical certification that is meaningful for evaluating policies because $\ell_{\alpha}(\mathcal{D})$ is informative about the outcome of, say, a future patient in a population. By contrast, using the methodology of Huang et al. one could provide a confidence interval for any given quantile in a loss distribution. However, this approach would provide a much less direct inference about the outcome of a future patient. We will clarify this point in the problem formulation.
>
> In future work, we plan to investigate extensions of our method to provide $(1-\delta)$-tolerance regions, i.e.,  $P_{\mathcal{D}}[  P_{\pi} \\{ L_{n+1} \leq \ell_{\alpha}(\mathcal{D}) |  \mathcal{D} \\}  \geq 1-\alpha  ] \geq 1 - \delta$.

---

> > ### Author Response · Authors · 2023-05-02
> > **Reply to Reviewer ggu1, part b)**
> >
> > >\#7: ”While $\bar{w}$ is being estimated through data splitting, wouldn't the optimization process is Eqn 15 make the choice of  $\bar{w}$ indirectly depend on both the partitions of the data? What is the intuition for introducing $\beta$ and what is the trade-off that  $\beta$ is balancing, e.g., why is $\beta=\alpha$ not adequate?”
> >
> > Note that in (15) we select *one* element in the set $\{ \bar{w}_{\beta}(\mathcal{D_0}) : 0 \< \beta \< \alpha \}$. However, each element in the set depends only on *one* partition, namely, $\mathcal{D_0}$.
> >
> > The intuition for introducing $w_{\beta}(\mathcal{D_0})$ is to provide a statistically valid upper bound on the ratio (5) where $\beta$ only specifies its confidence level. Note $\ell_{\alpha, \beta}$ bounds the loss with a coverage probability of at least $1-\alpha$ for *any* given value of $0 \< \beta \< \alpha$. Therefore (15) chooses the tightest achievable bound.
> >
> > We will make these remarks in the revised manuscript after Theorem 4.1.
> >
> > >\#8: ”An important factor that separates this work from prior methods (Chandak et al. (2021) and Huang et al. (20221) is that it can deal with settings where the observational data has unobserved confounding, and this is possible by controlling the odds ratio (Eqn 6). Is there any particular reason to consider the ratio of the odds in Eqn 6 instead of bounding $p_{\pi}(A|X) / p(A|X,U)$ directly? Empirically, it *seems* like a practitioner will have a better idea of the latter. Theoretically, could one not use these bounds in the works by Chandak et al. (2021) and Huang et al. (20221) to directly extend their bounds for the desired setting with unobserved confounders and unknown behavior policy? It feels like this would just affect the leading constant on the Hoeffding/DKW-like bound which currently depends on the max of the IPW/importance ratio's values.”
> >
> > An alternative to the specified sensitivity model (6) could indeed be to bound the ratio $\hat{p}(A|X) / p(A|X,U)$ directly. However, one cannot use a bound parameterized as \
> > $1/ \Gamma \leq \hat{p}(A|X) / p(A|X,U) \leq \Gamma$ \
> > but rather one must specify it in terms of two parameters: \
> > $\Gamma_1 \leq \hat{p}(A|X) / p(A|X,U) \leq \Gamma_2$ \
> > for the resulting probabilities to be bounded within 0 to 1. This makes it harder to specify the credibility of our assumptions on $\hat{p}(A|X)$.
> >
> > The sensitivity model that we consider is well established in the literature, see for example: Tan. “A distributional approach for causal inference using propensity scores” (2006), Zhao et. al. “Sensitivity analysis for inverse probability weighting estimators via the percentile bootstrap” (2019), Jin et al. “Conformal Sensitivity Analysis for Individual Treatment Effects” (2023).
> >
> > >\#9: ”I only got a chance to skim through the proof but I think as a reader it would be helpful if the authors can discuss intuitively what Eqn 13 and 15 are capturing as well. Here it will also be beneficial to rediscuss the setting where $\tau = 1$, similar to how it was done in Section 3. These might make the draft more readable and perhaps the proof can be deferred to the appendix.”
> >
> > We will add remarks after Theorem 4.1 that discuss eqs. (13) and (15). Setting $\Gamma=1$ leads to $\underline{W} = \overline{W}$ in (11), which we will point out in the revised manuscript.
> >
> > >\#10: ”I am intrigued by the non-monotonous nature of the curves in Figure 4 and 6. Can authors share insight on why it is not monotonous?”
> >
> > Please note that the figures show the miscoverage *gap*, which is not expected to have a monotonous form on theoretical grounds.
> >
> > >\#11: ”Can authors discuss how to leverage their proposed method when one only requires to get coverage over the expected value of the loss/reward for the evaluation policy? (I suspect this might not be possible and would connect back to point 6 discussed earlier. But maybe I am wrong, and perhaps authors can correct me over here. )”
> >
> > To infer the expected value $\mathbb{E}[L_{n+1}]$, rather than the actual loss $L_{n+1}$, requires computing a valid confidence interval which we do not consider in our paper. (See e.g. Huang et al.)
> >
> > >**Broader Impact Concerns:**  Since this paper deals with topics that aim at safety and credibility, it might benefit from more discussion on assumption implied by Eqn 6 on the bounds of the odds ratio. How should this be chosen in practice, etc.?
> >
> > As we argue in the paragraph above Figure 2 – and illustrate in the examples – one should *not* consider a single $\Gamma$, but rather policy evaluation should be performed across a range of plausible odds bounds $\Gamma$ which correspond to increasing credibility of our assumptions on $\hat{p}(A|X)$.

---

### Review · Reviewer_jTUo · 2023-04-03

**Summary Of Contributions:**

The authors study the problem of off-policy evaluation, where we want to evaluate the performance of a target policy with only past observational data. The authors leverage recent advances in conformal predictions to obtain finite-sample coverage guarantees over the entire loss distribution. Due to the benefits of conformal prediction techniques, the new approach allows to take model misspecifications into account. The authors validate their methods on both synthetic data and real world dataset, and can obtain relatively reasonable coverage and out-of-sample guarantees.

**Audience:**

Yes

**Claims And Evidence:**

Yes

**Requested Changes:**

- Please add a discussion with your method and some classical approaches such as utilizing confidence bounds to obtain confidence intervals, or some distributional robust optimization based approach (e.g., Empirical likelihood for contextual bandits).

- Please add some more real-world experiments.

- Since your method requires data spliting, can you comment or discuss the advantage and disadvantage of splitting based methods under the OPE setting? e.g., you can not leverage all the samples to construct the upper and lower bound?


- Please add a related work session and move the proof part to the appendix.

- please distinguish citet and citep.


**Strengths And Weaknesses:**

**Strengths**

- The split method is easy to implement, and alleviates the strong assumption which is required by previous methods.

- As stated previously, the assumption here is relatively mild, which is good for real-world applications.

- The proof written in the paper is easy to follow.


**Weaknesses**

- There is not much baselines to compare in the experiment part, and also I haven't see any discussion with previous work or standard approaches.

- Still, the experiments part is a little bit weak. There is not much more real experiments.

---

> ### Author Response · Authors · 2023-05-02
> **Reply to Reviewer jTUo**
>
> We thank the reviewer for the feedback and provide our responses below.
>
> > "Please add a discussion with your method and some classical approaches such as utilizing confidence bounds to obtain confidence intervals, or some distributional robust optimization based approach (e.g., Empirical likelihood for contextual bandits)."
>
> In contrast to using confidence intervals for any given quantile in a loss distribution, we consider a direct inference about the out-of-sample loss of, say, a future patient. We will clarify this point in the problem formulation. The contrast with previous off-policy evaluation methods is highlighted at the very end of Section 3.
>
> > "Please add some more real-world experiments."
>
> We have performed off-policy evaluation using data from the International Human Dimensions Programme (IHDP) which has been analyzed in the causal inference literature. We will add these results to the manuscript.
>
> > "Since your method requires data splitting, can you comment or discuss the advantage and disadvantage of splitting based methods under the OPE setting? e.g., you can not leverage all the samples to construct the upper and lower bound?"
>
> In general, data splitting ensures statistically valid inferences at the expense of efficiency. In the OPE-setting this means that the limit curves are valid in a wide range of circumstances but possibly conservative.
>
> > "Please add a related work session and move the proof part to the appendix."
>
> We believe that Section 3 provides a fairly comprehensive background of related work. We agree with the reviewer that the proof can be moved to an appendix, and instead use the space to elaborate the description of the method.
>
> > "please distinguish citet and citep."
>
> Thank you for bringing this to our attention. We have made some updates in the text.

---

### Review · Reviewer_s42E · 2023-04-12

**Summary Of Contributions:**

This paper studies the problem of off-policy evaluation, i.e., evaluating the performance of a decision policy using past data. The authors provide results for the entire loss distribution instead of the expected value of policies, i.e., Eq. (2).

Section 3 summarizes existing baseline methods. There exist estimators of the mean, as noted in Eqs. (7) and (8), and the doubly robust estimator, and similar estimators can be generalized to estimate entire loss distributions, as shown in and after Eq. (9).

Section 4 introduces the methods based on splitting the dataset and the proposed estimator in Eq. (12), and presents the main results, i.e., the limit in Eq. (15) satisfies Eq. (2). The authors then give the implementation (Algorithm 1) and prove Theorem 4.1.

Section 5 contains experiments. The authors first define several metrics, including informativeness (Eq. (23)), miscoverage gap (Eq. (24)). They use synthetic data and real data from the the National Health and Nutrition Examination Survey (NHANES). The results show that the informativeness of most curves stays a high level (about 90%), and the proposed method is slightly conservative, but remains valid for all $\alpha$ values.

**Audience:**

Yes

**Broader Impact Concerns:**

The experiments are conduced using synthetic data and public dataset. I do not see concerns on the ethical implications of the work.

**Claims And Evidence:**

Yes

**Requested Changes:**

First, I get that the metric of entire loss of distribution is different with the expected loss. However, I did not get how the entire loss of distribution is useful in off-policy evaluation. The authors mention that in the introduction, when the distribution of losses is skewed or is widely dispersed, expected loss itself is limited. I am wondering show the entire loss distribution could be used after off-policy evaluation. Some concrete examples of showing that the entire loss distribution could provide a different solution that using expected loss would be great.

Second, the baseline methods are well discussed in Section 3, since they look closely related to existing mean estimators. However, in Section 4, the proposed method is presented without much intuitions or discussion. The authors mention that in Section 3 "this estimator is
consistent, it is not guaranteed to yield a proper cdf". I would like to see more comments on why the authors design the proposed estimator in such a way and how intuitively the proposed methods could resolve issues of existing methods.

Third, the proposed method does not change over different values of $\alpha$ in Figures 4 and 6. Could you comment on this.

**Strengths And Weaknesses:**

**Strengths**

The introduction of the problem and motivation is clear. The problem of study is relevant.

Measuring the entire loss distribution seems novel to me, since most existing results are for expected values.

The proposed methods are with guarantees and supported by experimental results.

**Weaknesses**

The usefulness of the new metric (entire loss distribution) in off-policy evaluation is not well explained.

The authors do not provide explanations for the proposed methods.

---

> ### Author Response · Authors · 2023-05-02
> **Reply to Reviewer s42E**
>
> We thank the reviewer for the comments and provide our in-line responses below.
>
> > "First, I get that the metric of entire loss of distribution is different with the expected loss. However, I did not get how the entire loss of distribution is useful in off-policy evaluation. The authors mention that in the introduction, when the distribution of losses is skewed or is widely dispersed, expected loss itself is limited. I am wondering show the entire loss distribution could be used after off-policy evaluation. Some concrete examples of showing that the entire loss distribution could provide a different solution that using expected loss would be great."
>
> Characterizing the entire loss distribution of a policy provides more information than its expected loss. For instance, consider a case similar to Figure 7 (Right): Suppose the expected loss of high seafood consumption for women was 7$\mu$g/L, then this consumption policy would be deemed safe. However, this figure masks the tail losses that a substantial proportion of women may face under such a policy! This would be an instance where the expected loss in a population may be small, but the tail losses can be unacceptably large.
>
> This is a motivating case stated in the introduction of the paper, and we will return to the point in the experimental results in the revised manuscript.
>
> > "Second, the baseline methods are well discussed in Section 3, since they look closely related to existing mean estimators. However, in Section 4, the proposed method is presented without much intuitions or discussion. The authors mention that in Section 3 "this estimator is consistent, it is not guaranteed to yield a proper cdf". I would like to see more comments on why the authors design the proposed estimator in such a way and how intuitively the proposed methods could resolve issues of existing methods."
>
> We agree with the reviewer that the description of the method in Section 4 could be improved by elaborating on its properties immediately after the result. We will do so in the revised manuscript.
>
> Please note that in contrast to the methods in Section 3 we are interested in providing a predictive limit curve with out-of-sample guarantees.
>
> > "Third, the proposed method does not change over different values of $\alpha$ in Figures 4 and 6. Could you comment on this."
>
> The horizontal line merely indicates the zero-level, not the performance of the method. We will revise the plot style for clarity.

---

### Review · Reviewer_N8A4 · 2023-04-14

**Summary Of Contributions:**

The paper proposes a method for building prediction bounds on out-of-sample loss when a new policy is applied to a given population. The method aims to provide finite-sample guarantees even under model misspecification and unobserved confounding, under assumptions of varying strength under a marginal sensitivity model. The method makes use of a conformal argument combined with data splitting, and a proof is given that combines elements from two previous works on conformal arguments under distributon shift, and on bounding ITEs under confounding. The method is shown to be valid in two simulated settings; one with a well-specified, known propensity model, and one with unobserved confounding and an estimated propensity model. The method is then demonstrated on NHANES data regarding mercury levels and shellfish consumption.

**Audience:**

Yes

**Broader Impact Concerns:**

No major concerns.

**Claims And Evidence:**

Yes

**Requested Changes:**

There are copied from above:

 - A description of what the method does when $\Gamma = 1$, with a comparison to the IPW baseline. This would give some intuition about the core approach before we insert bounds on importance weights.
 - Some intuition about the quantities $\beta$ and $\bar w_\beta(\mathcal D_0)$, and what their role is.
 - Some commentary, and refinement of claims, regarding finite sample statistical uncertainty, in both the estimation of $\hat p(A | X)$ and $\hat F$.
 - Clarifications to other minor points, including double-checking notation in the proof.

**Strengths And Weaknesses:**

**Strenghts:**

The paper addresses an important topic, and the idea of deriving limit curves, rather than bounds on expected policy value, is interesting. The method seems to be a useful contribution and appears to work well. There is a rigorous justification for the method (although I could use some clarification).

**Weaknesses**

**W1:** I did not come away from the paper with a solid understanding of the role of each piece of the method. It would be useful to introduce the each part of the method and the role that they play in isolation before combining them together. In this respect, the marginal sensitivity model is well-motivated, but the limiting technique involving $\beta$ is far less clear. I would be interested to see what the method looks like when the propensity score is assumed to be known and well-specified, and to have a better understanding of how the method differs from IPW in that case. Perhaps this was covered by previous work, but if so, it would be useful to have a short description in this paper.

**W2:** It seems that the assumptions of the marginal sensitivity model are used in this work to handle both statistical uncertainty in estimating a model and misspecification/unobserved confounding. The former seems to be both well-characterized by previous asymptotic and finite-sample concentration arguments, and not especially amenable to the marginal sensitivity model framing where all errors $\hat p(A \mid X)$ need to be bounded probability 1. By contrast, most statistical uncertainty bounds are presented wrt some probability $\delta$. How do the authors suggest reasoning about statistical uncertainty when choosing $\Gamma$? Should $\Gamma$ be chosen differently at different sample sizes? This would run contrary to advice in other parts of the causal inference literature, where $\Gamma$ is considered purely a population quantity, rather than also quantifying assumptions about finite-sample statistical uncertainty.

**W3**: The authors claim that the method handles finite sample uncertainty, but it seems like there is still uncertainty that is not quantified here. Setting aside the uncertainty in estimating $\hat p(A | X)$ discussed above, there is also uncertainty in estimating $\hat F$ that does not seem to be discussed (for example, in the proof, IIUC, arguments are made wrt the expectation of $\hat F$). Could the authors clarify that this paper provides, essentially, a point estimate of the limit curves, but that there is additional statistical uncertainty in these limit curves? Relatedly, could uniform confidence bounds on CDFs, like the DKW bounds, be applied to obtain inference about the limit curves?

**W4**: The proof could be made clearer, especially regarding the probability measures with respect to which the probabilities of events and expectations are being defined. See some of the minor comments below.

**Minor Comments**

The role of $\beta$ in Theorem 4.1 is not clear from the statement of the theorem. Could you provide some more intuition about it?

Page 5: Could (12) be called something other than a “proxy” for the CDF? It’s essentially a lower-confidence bound on the CDF, correct?

What is the interpretation of $\beta$ in Theorem 4.1?

On page 6 it is claimed that $p(\mathcal D_+)w_{n+1} = p(\mathcal S_+)w_{n+1}$. This doesn’t seem right to me. Where does the combinatorial factor corresponding to permutations go?

I’m a little confused about the probability measures wrt which many of the event probabilities are defined in Theorem 4.1. For example, in the display before (16), is the probability measure $\mathbb P\{L_{n+1} = \ell_i \mid \mathcal S_+\}$ defined with respect to $p$ or $p_\pi$? It seems like it should be $\mathbb P_\pi$? Likewise, should the probability in the the event in (16) be wrt $\mathbb P_\pi$ not $\mathbb P$? After eq (17), there is a $\mathbb P_\pi$ inside they expectation, but what measure is the outer expectation taken over?

Should the limit in (17) be a function of both $\alpha$ and $\beta$? I.e., should it be written $\ell_{\alpha, \beta}(\mathcal D_1, \bar W_{n+1})$?

It seems strange in (28) to define the ground truth as a function of the estimated nominal propensity score. It would seem more realistic to define the estimation procedure such that $\hat p(A =0 | x)$ concentrates appropriately to satisfy the $\gamma$ bound with high probability.

There should probably be a sentence that says “informativeness” corresponds to the right limit of the curves in Figures 3, 5 (I think). It was difficult for me to interpret at first.

It would be useful to have a clear description of how the proposed method differs from the IPW method in the correctly specified case.

It would be easier to interpret the takeaway from the NHANES experiment if the high-consumption rate of shellfish under the current policy were also given. I.e., if high consumption is relatively rare, then it makes sense that the rate of exceeding the guideline level will be relatively low (closer to $\pi_0$) under the current policy.

---

> ### Author Response · Authors · 2023-05-03
> **Our reply, part a)**
>
> We thank the reviewer for the detailed feedback and suggestions, they have been helpful in improving our work.
>
> > W1: "I would be interested to see what the method looks like when the propensity score is assumed to be known and well-specified, and to have a better understanding of how the method differs from IPW in that case."
>
> > 1: "A description of what the method does when $\Gamma = 1$, with a comparison to the IPW baseline. This would give some intuition about the core approach before we insert bounds on importance weights."
>
> We did in fact consider this in Section 5.1 Case: Known past policy ($\Gamma=1$), where we used the IPW-based quantile in (25) as a benchmark. In this case, we have that $\underline{W} = \overline{W}$ in (12). This leads to a slightly more conservative limit curve but remains on the other hand valid across $\alpha$, as seen in Figure 4.
>
> We agree with the reviewer that the description of the method in Section 4 could be improved by elaborating on its properties immediately after the result. We will do so in the revised manuscript.
>
> > W2: "It seems that the assumptions of the marginal sensitivity model are used in this work to handle both statistical uncertainty in estimating a model and misspecification/unobserved confounding. The former seems to be both well-characterized by previous asymptotic and finite-sample concentration arguments, and not especially amenable to the marginal sensitivity model framing where all errors $\hat{p}(A|X)$ need to be bounded probability 1. By contrast, most statistical uncertainty bounds are presented wrt some probability $\delta$. How do the authors suggest reasoning about statistical uncertainty when choosing $\Gamma$? Should $\Gamma$ be chosen differently at different sample sizes? This would run contrary to advice in other parts of the causal inference literature, where $\Gamma$ is considered purely a population quantity, rather than also quantifying assumptions about finite-sample statistical uncertainty."
>
> As we argue in our paper (paragraph above Figure 2), we do not advice using a single $\Gamma$, but rather policy evaluation should be performed across a range of plausible odds bounds $\Gamma$ which correspond to increasing credibility of our assumptions on $\hat{p}(A|X)$. The upper limit to this range is when the resulting limit curve is no longer deemed informative for the problem at hand.
>
> The model in (6) can accommodate all possible sources of errors in the probability model $\hat{p}(A|X)$ – both model misspecification *and* finite-sample errors. Note that any such model errors will be bounded by 1 and therefore (6) is applicable. In the unlikely event of using a perfectly specified model – with no unmeasured confounding – then indeed the range of $\Gamma$ can be narrowed towards 1 as the sample size increases. However, since the possibility of unmeasured confounding always looms large, we advice against using such incredible model assumptions.
>
> > 2: "Some intuition about the quantities $\beta$ and $\bar{w}_{\beta}(\mathcal{D}_0)$, and what their role is."
>
> The intuition for introducing $\bar{w}_{\beta}(\mathcal{D_0})$ is to provide a statistically valid upper bound on the ratio (5) where $\beta$ only specifies its confidence level. We will clarify this point in the revised manuscript after Theorem 4.1.
>
> > W3: "The authors claim that the method handles finite sample uncertainty, but it seems like there is still uncertainty that is not quantified here. Setting aside the uncertainty in estimating $\hat{p}(A|X)$ discussed above, there is also uncertainty in estimating $\hat{F}$ that does not seem to be discussed (for example, in the proof, IIUC, arguments are made wrt the expectation of $\hat{F}$). Could the authors clarify that this paper provides, essentially, a point estimate of the limit curves, but that there is additional statistical uncertainty in these limit curves? Relatedly, could uniform confidence bounds on CDFs, like the DKW bounds, be applied to obtain inference about the limit curves?"
>
> > 3: "Some commentary, and refinement of claims, regarding finite sample statistical uncertainty, in both the estimation of $\hat{p}(A|X)$ and $\hat{F}$."
>
> Please note that $\hat{F}$ in (12) is not considered as an estimate of the CDF. Rather, it is merely a device to construct a predictive limit on $L_{n+1}$ which is the interest of this paper.
>
> Regarding the uncertainty in $\hat{p}(A|X)$, see the reply above.

---

> > ### Author Response · Authors · 2023-05-03
> > **Our reply, part b)**
> >
> > **Minor Comments**
> > > "The role of $\beta$ in Theorem 4.1 is not clear from the statement of the theorem. Could you provide some more intuition about it?"
> >
> > Please see the reply above.
> >
> > > "Page 5: Could (12) be called something other than a “proxy” for the CDF? It’s essentially a lower-confidence bound on the CDF, correct?"
> >
> > We simply called it a ‘proxy’ to avoid mistaking it for a proper CDF-estimator, which is not of interest in this paper.
> >
> > > "On page 6 it is claimed that $p(\mathcal{D_{+}})w_{n+1} = p(\mathcal{S_{+}})w_{n+1}$. This doesn’t seem right to me. Where does the combinatorial factor corresponding to permutations go?"
> >
> > Due to the weight $w_{n+1}$, the distribution $p(\mathcal{D_{+}})$ is taken over a sequence of iid samples. Therefore it is invariant to any permutation $\mathcal{S_{+}}$ of the samples.
> >
> > > "I’m a little confused about the probability measures wrt which many of the event probabilities are defined in Theorem 4.1. For example, in the display before (16), is the probability measure $\mathbb{P} \\{ L_{n+1} = \ell_i | \mathcal S_+ \\}$ defined with respect to $p$ or $p_{\pi}$? It seems like it should be $\mathbb P_{\pi}$? Likewise, should the probability in the the event in (16) be wrt $\mathbb P_{\pi}$ not $\mathbb{P}$? After eq (17), there is a
> > $\mathbb{P}_{\pi}$ inside they expectation, but what measure is the outer expectation taken over?"
> >
> > The choice of notation follows that of Tibshirani et al. “Conformal Prediction Under Covariate Shift” (2019). While convenient for covering samples drawn from two different distributions – $P_\pi$ and $P$ – it has the potential drawback of being unclear at times. We will clarify the usage with more detail in the proof when ambiguities may arise.
> >
> > > "Should the limit in (17) be a function of both $\alpha$ and $\beta$? I.e., should it be written $\ell_{\alpha, \beta}(\mathcal D_1 ,  \bar W_{n + 1})$?"
> >
> > Yes, the limit in (17) does depend on both $\alpha$ and $\beta$, but as we show a few lines later, $\beta$ plays no role in determining the miscoverage since it is canceled out by introducing $w_{\beta}$.
> >
> > > "It seems strange in (28) to define the ground truth as a function of the estimated nominal propensity score. It would seem more realistic to define the estimation procedure such that $\hat{p}(A=0|x)$ concentrates appropriately to satisfy the $\gamma$ bound with high probability."
> >
> > We agree that it may appear unnatural to define the true unknown policy as in (28), but we are here mainly interested in accurately controlling its divergence from the nominal model by following the approach of Jin et al. (2023).
> >
> > > "There should probably be a sentence that says “informativeness” corresponds to the right limit of the curves in Figures 3, 5 (I think). It was difficult for me to interpret at first."
> >
> > Yes, we will clarify it in the text.
> >
> > > "It would be useful to have a clear description of how the proposed method differs from the IPW method in the correctly specified case."
> >
> > Even in the well-specified case, the limit curve obtained by the IPW method offers no valid out-of-sample guarantees of the evaluated policy. We do in fact provide this contrast in Figure 4 in Section 5.1.
> >
> > > "It would be easier to interpret the takeaway from the NHANES experiment if the high-consumption rate of shellfish under the current policy were also given. I.e., if high consumption is relatively rare, then it makes sense that the rate of exceeding the guideline level will be relatively low (closer to $\pi_{0}$) under the current policy."
> >
> > We will add the figure for the high-consumption rate in the revised manuscript.

---

### Decision · Action_Editors · 2023-06-21

**Recommendation:** Accept with minor revision

**Comment:**

Following the reviewers' recommendations, the paper is worth publication in TMLR. However, one of the reviewers asked for clarification on the primary sensitivity parameter introduced by the authors. Before acceptance, the authors should address their concerns or at least add some details to the paper.


**Audience:**

The problem that the paper focus on is an important problem and the method developed there should be of interest for the community.

**Claims And Evidence:**

The authors of the paper address the problem of off-policy evaluation, which involves evaluating the relevance of a target policy using past observations. To ensure coverage across the entire loss distribution, the authors leverage recent advancements in conformal predictions. This innovative method also allows for the incorporation of model misspecifications. By validating their approaches using synthetic data and real-world datasets, the authors demonstrate satisfactory coverage and out-of-sample guarantees.

---

> ### Author Response · Authors · 2023-07-16
> **Revision**
>
> Dear AE,
> We have uploaded the revised paper that addresses the question of choosing the sensitivity parameter Gamma (specifically Sec. 2 and real-data examples in Sec 5.2 + A.2).
>
> We hope these revisions will clarify that Gamma is to be chosen in a range that yields informative limit curves.
>
> sincerely,
> Dave Zachariah